# Social Networking Site Use and Emotional Eating Behaviors among Chinese Adolescents: The Effects of Negative Social Comparisons and Perspective-Taking

**DOI:** 10.3390/bs13090768

**Published:** 2023-09-14

**Authors:** Shan Sun, Siying Chen, Zian Wang, Yan Xiong, Shixuan Xie

**Affiliations:** Department of Psychology, Hubei University, Wuhan 430062, China

**Keywords:** emotional eating, social networking site, negative social comparisons, perspective taking

## Abstract

Emotional eating has emerged as a significant disordered eating and public health concern among adolescents. Despite the widespread prevalence of social networking site (SNS) use among this population, research investigating the influence of SNS use on adolescent eating behaviors remains limited. This study is to examine the impact of SNS use on emotional eating among Chinese adolescents, with a specific focus on exploring the mediating role of negative social comparisons and the moderating effect of perspective-taking. Data were obtained through an online survey involving 778 middle school students in China. The findings indicate that SNS use exerts a positive influence on adolescents’ engagement in emotional eating, with this association being mediated by the presence of negative social comparisons. Perspective-taking demonstrated a protective role in the context of adolescents’ utilization of social media platforms. For individuals characterized by high levels of perspective-taking, the effects of SNS use on negative social comparisons are mitigated, subsequently reducing its impact on emotional eating.

## 1. Introduction

Emotional eating is commonly defined as the urge to eat in response to negative emotions, without physiological hunger [1,2,3]. It is considered maladaptive eating behavior associated with a range of negative physiological and psychological health outcomes [4,5]. Emotional eating is associated with weight gain [6]; increased consumption of high-fat, sweet, and energy-rich food [7]; as well as psychiatric symptoms [8]. Throughout the COVID-19 pandemic, there has been a notable increase in the prevalence of emotional eating, which refers to the consumption of food as a mechanism for regulating negative emotions. Empirical investigations conducted in China and various other regions have documented a prevailing occurrence rate ranging from 17% to 20% [9,10]. Adolescence is a developmental stage characterized by simultaneous biological, social, and psychological changes. During this period, identity exploration, self-expression, the formation of friendships, and peer acceptance hold significant importance for adolescents [11]. Adolescence has been identified as a critical period for developing eating disorders [12]. Various factors can influence emotional eating, and the primary influencing factors for emotional eating in adolescence include biological factors [13], psychological factors [14,15], sociocultural factors [16,17], and family factors [18]. With the widespread use of social media among adolescents, the influence of social media on their eating behavior has become increasingly significant [19]. This study examines the effects of adolescent social networking site (SNS) use on emotional eating and explores the underlying psychological mechanisms.

Adolescence is a critical period for the occurrence of emotional eating, influenced by the interaction of physiological changes and environmental factors [13,20]. Previous theories have suggested that individuals engage in emotional eating as a coping strategy to alleviate emotional distress [21]. However, the recently developed affect regulation model proposes that an increase in negative emotions triggers emotional eating, as individuals attempt to temporarily distract their attention, numb their emotions, or seek comfort [22]. In addition to coping with negative emotions, the escape theory suggests that excessive eating as a result of dealing with negative emotions is an attempt to escape or shift attention away from self-threatening stimuli, as these stimuli evoke self-disgust and negative affect [23]. Whether aiming to cope with negative emotions or self-awareness, emotional eating only provides temporary relief and does not effectively eliminate the underlying threats faced by individuals.

Social media serves as an information platform for adolescents to acquire knowledge, engage in communication, and express themselves, encompassing various functions such as learning, leisure, and entertainment. With the widespread use of social networking site, researchers have begun to investigate the influence of social networking site use on eating behaviors among adolescents. Multiple studies have found a clear relationship between social networking site usage and patterns of eating behavior among adolescents [24,25]. Through a survey conducted on 996 Australian adolescents in Grade 7 and 8, Wilksch et al. found significant correlations between the usage of Facebook, Instagram, Snapchat, and Tumblr and disordered eating behaviors among the participants [24]. Another study involving 383 college students discovered that lower weight and appearance esteem acted as mediators in the connection linking excessive social networking site use and restrained eating behaviors for both genders [25]. Furthermore, two meta-analytic studies emphatically indicated a significant association between social networking site use and the risk of disordered eating behaviors [26,27]. In addition, teenagers not only share their emotional experiences on social media platforms but are also influenced by emotional content found on these platforms. Relevant research has found that social networking site use may lead to negative emotions such as anxiety and depression among adolescents [28], as well as unfavorable social adaptation outcomes [29]. The negative emotions and stimuli experienced on social media platforms are likely to trigger adolescents’ emotional eating behaviors [30].

In today’s information society, social networking sites have become important venues for triggering social comparisons [31]. As significant platforms for self-presentation and expression, individuals not only post a substantial amount of self-relevant information on social networking sites but also unavoidably become recipients of information posted by other users [32]. Moreover, social comparison is an unconscious and spontaneous process, where exposure to others’ information triggers individuals’ social comparisons [33]. Furthermore, as an idealized platform for self-presentation, individuals exercise more control and strategic behavior in presenting themselves on social networking sites. They tend to present and publish information with a positive bias, emphasizing their positive aspects and enjoyable experiences, such as showcasing attractive selfies and pleasant life experiences [26]. Social comparison theory suggests that exposure to positively biased information leads to negative self-perceptions and evaluations, where individuals perceive others as better off and happier than themselves, resulting in negative social comparisons [34]. Researchers define this negative self-perception and evaluation caused by social comparison as negative social comparison and believe that it has severe negative effects on individuals’ psychological and social adjustment [35].

Negative social comparisons among adolescents on social networking sites often lead to unfavorable outcomes [36]. Engaging in negative social comparisons and interactions on these platforms can result in elevated levels of depression and anxiety [37,38]. In a study conducted by Wu et al., targeting 997 Chinese university students, it was observed that utilizing WeChat Moments for negative social comparisons—perceiving oneself as inferior to others and recognizing others’ lives as more fulfilling than one’s own—triggered depressive emotions [37]. Furthermore, investigations involving 511 Chinese secondary school students revealed that when adolescents engage in appearance-based comparisons on social networking sites, it triggers anxiety about their own physical appearance, subsequently leading to the adoption of emotionally driven eating as a coping mechanism in response to such negative affect [19]. Consequently, negative social comparisons within the realm of social networking sites emerge as a crucial factor influencing individuals’ dietary patterns. Building upon the aforementioned findings, the present study hypothesizes that negative social comparison plays a mediation role in the association between social networking site usage and emotional eating.

The process of adopting or imagining the viewpoints of others, commonly referred to as perspective-taking [39], encompasses the cognitive ability to perceive and interpret the world through the lens of another individual [40]. Additionally, it involves the psychological processes through which individuals, stemming from their own standpoint or situational context, engage in the mental simulation or estimation of others’ perspectives and attitudes [41,42]. It is worth noting that perspective-taking not only influences the way in which individuals process external social information but also impacts their self-perception. Through engaging in perspective-taking, individuals can enhance their perception of shared attributes between themselves and others [43], or alternatively, blur the boundaries that distinguish oneself from others [44]. The mediating role of self–other overlap has been identified as a significant factor in the relationship between perspective-taking and changes in self-concept [45]. For instance, when individuals engage in perspective-taking towards others, their self-perception tends to undergo alterations, leading them to perceive a greater similarity between themselves and the individuals whose perspectives they have considered. Furthermore, Galinsky et al. [40] discovered that participants who underwent perspective-taking exercises focused on elderly individuals exhibited behavioral patterns that were more closely aligned with those of the elderly themselves. Similarly, completing visual-spatial perspective-taking tasks has been shown to significantly enhance individuals’ perception of self–other similarity [46]. This heightened sense of perceived similarity may contribute to a decrease in negative social comparisons. In addition, researchers have explored the influence of individuals’ self-esteem levels on social comparisons. Individuals with low self-esteem not only engage in more negative social comparisons on social networks [47] but also derive more negative feelings from these comparisons [48]. However, perspective-taking is believed to maintain a positive association with self-esteem. Consequently, the increased self-esteem resulting from the process of perspective-taking may serve to mitigate negative social comparisons within the realm of social networking platforms. Based on these considerations, the present study hypothesizes that perspective-taking plays a moderating role in the relationship between social networking site usage and negative social comparisons.

The main purpose of this study is to examine the relationship between adolescent social media use, negative social comparison, and emotional eating, while also investigating the moderating role of perspective-taking. The hypotheses are as follows: (1) There is a significant positive correlation between social networking site use and emotional eating. (2) There is a significant positive correlation between social networking site use and negative social comparison during social networking site use. (3) There is a significant positive correlation between negative social comparison and emotional eating. (4) Negative social comparison mediates the relationship between social networking site use and emotional eating. (5) Perspective-taking moderates the effect of social networking site use on negative social comparison; specifically, for individuals with high levels of perspective-taking, the influence of social networking site use on negative social comparison is weaker.

## 2. Method

### 2.1. Participants

The present study utilized a sample of 815 middle school students who were recruited from Guangdong Province, China. Data were collected using snowball sampling on online social media platforms (www.wjx.com, accessed on 30 June 2023), which is similar to Qualtrics) used by high school students. Among the participants, a total of 778 individuals provided valid questionnaires, with cases involving missing or irregular data excluded from analysis. The response rate for completing the survey was determined to be 95.46%. Within the remaining sample, there were 368 female participants and 410 male participants. The average age of the participants was 13.01 years (SD = 0.49). Additionally, the mean body mass index (BMI) of the participants was 24.97 (SD = 6.16). The study was approved by the ethics committee of Normal School of Hubei University (approval code: HBU-NS-202306011). Adolescents were informed about the study, and written consent was obtained from those who agreed to participate.

### 2.2. Measures

#### 2.2.1. Demographics

The participants were requested to provide demographic information, which encompassed gender, age, and measurements of height and weight for the purpose of calculating body mass index (BMI, kg/m^2^).

#### 2.2.2. Social Networking Site Use

Social networking use was assessed using the Social Networking Usage Intensity Scale developed by Ellison, Steinfield, and Lampe [49] and translated by Niu et al. [29]. The questionnaire comprised a total of eight items, with the first two items measuring individuals’ friend count on social media platforms and their average daily usage time. The remaining six items were designed using a 5-point Likert scale (ranging from 1, “strongly disagree”, to 5, “strongly agree”) to evaluate the emotional attachment to social networking sites and the extent to which social media integrated into individuals’ lives (e.g., “WeChat moment is part of my everyday activity”). Participants were instructed to respond to the questionnaire based on their actual engagement with WeChat Moments. WeChat Moments is a popular social networking platform widely used by Chinese teenagers. It allows users to share photos, videos, and written posts with their contacts. It serves as a digital diary for users to document their daily experiences and connect with friends. In the current study, the Cronbach’s alpha coefficient for this questionnaire was 0.87.

#### 2.2.3. Emotional Eating

The measurement of emotional eating was conducted using three items from the revised 18-item Three-Factor Eating Questionnaire (TFEQ-R18) [50], consistent with previous research [51]. The TFEQ-R18 was derived from factor analyses of the original 51-item TFEQ in a sizable sample of obese individuals [50] and has been demonstrated to be applicable to the general population [52]. Moreover, satisfactory internal consistency of the TFEQ-R18 has been observed in measurements involving Chinese participants [53]. The items included in the measurement were as follows: “When I feel anxious, I find myself eating”, “When I feel blue, I often overeat”, and “When I feel lonely, I console myself by eating”. Participants were required to indicate their level of agreement on a six-point Likert scale, ranging from “strongly disagree (1)” to “strongly agree (6)” for each item. In the current study, the Cronbach’s alpha coefficient for this questionnaire was 0.90.

#### 2.2.4. Negative Social Comparison

The Negative Social Comparison Scale, developed by Lee [54] and translated by Wu et al. [37], was employed to measure the extent to which individuals engage in negative social comparison (perceiving others as better than oneself) on social media platforms. The questionnaire consisted of three items, rated on a 5-point Likert scale (ranging from 1, “strongly disagree”, to 5, “strongly agree”). Participants were instructed to respond to the questionnaire based on their actual usage of social networking sites (specifically WeChat Moments). Higher scores indicated a higher degree of engaging in negative comparisons on social media platforms. In the current study, the Cronbach’s alpha coefficient for this questionnaire was 0.88.

#### 2.2.5. Perspective-Taking

Perspective-taking was assessed using four items adapted from the research conducted by Grant et al. [55] and derived from the perspective-taking measure developed by Davis et al. [56]. The selected items for measurement were: “I look at things from the perspective of others”, “I imagine how my actions will affect things that are important to others”, “I understand why particular issues hold emotional significance for others”, and “I look at matters in terms of other people’s personal concerns”. Participants were instructed to rate their level of agreement on a six-point Likert scale, ranging from “strongly disagree (1)” to “strongly agree (6)” for each item. In the current study, the Cronbach’s alpha coefficient for this questionnaire was 0.92.

## 3. Results

### 3.1. Common Method Bias Test

In a cross-sectional study, it is essential to consider the potential occurrence of common method bias, which can introduce systematic errors. One widely used test to evaluate this bias is Harman’s single factor test [57]. Applying Harman’s single factor test in our study revealed that only 30.77% of the variance was explained by a single factor, which is below the accepted cut-off value of 50%. This suggests that common method bias is not a major concern in our study. To further examine common method bias, we utilized the single unmeasured latent method [57]. This method involved constructing a confirmatory factor analysis (CFA) model for convergent validation. The CFA model demonstrated acceptable fit indices: χ^2^/df = 3.32, TLI = 0.96, CFI = 0.97, RMSEA = 0.06. Next, an additional method factor was added to the original CFA model, with all measurement items loading on both their respective construct factors and the method factor. The results indicated that compared to the original model, the changes in all fit indices were less than 0.02, suggesting that the model did not significantly improve after incorporating the common method factor. Therefore, the results indicate that there is no significant evidence of common method bias in the measurements. Additionally, we examined the presence of non-response bias through comparing the first and last values of the variables in the dataset. However, we found that the relationship between these values was statistically insignificant. Consequently, we can conclude that there is no evidence of non-response bias in our study.

### 3.2. Correlations among Variables

The results of the correlation analysis between variables in the study are presented in Table 1. As hypothesized, a significant positive correlation was found between social networking use and emotional eating (r = 0.21, *p* < 0.01), indicating that increased social networking use was associated with higher levels of emotional eating. Furthermore, a significant positive correlation was observed between social networking use and negative social comparison (r = 0.50, *p* < 0.01), suggesting that individuals who used social networking more frequently were more likely to engage in negative social comparison.

Additionally, a significant positive correlation was found between negative social comparison and emotional eating (r = 0.23, *p* < 0.001), indicating that individuals who experienced higher levels of negative social comparison also tended to engage in more frequent episodes of emotional eating.

Furthermore, perspective-taking was found to have a significant positive correlation with social networking use (r = 0.11, *p* < 0.01) and emotional eating (r = 0.11, *p* < 0.01). However, a significant negative correlation was found between perspective-taking and negative social comparison (r = −0.09, *p* < 0.05).

### 3.3. Mediation Effect Test

This study employed the bias-corrected percentile bootstrap method from the SPSS macro PROCESS developed by Hayes [58] to test the core hypothesis of this study: whether negative social comparison mediates the relationship between social networking use and emotional eating. When using this method, Model 4 was selected, with 5000 iterations of resampling performed to compute a 95% confidence interval.

The results, as shown in Table 2, indicated a significant relationship between social networking use and emotional eating when controlling for gender, age, and BMI, with B = 0.11, SE = 0.02, and *p* < 0.001. Additionally, a significant relationship was found between emotional eating and negative social comparison, with B = 0.25, SE = 0.02, and *p* < 0.001. After including negative social comparison in the model, the relationship between social networking use and emotional eating remained significant, albeit with a weakened effect, with B = 0.06, SE = 0.02, and *p* < 0.01 (see Figure 1). Further analysis revealed a significant indirect effect of social networking use on emotional eating through negative social comparison, with an indirect effect of 0.05, SE = 0.01, 95% CI = [0.03, 0.08], and P_M_ = 43.50%. Additionally, the direct effect of social networking use on emotional eating was also significant, with a direct effect of 0.06, SE = 0.02, and 95% CI = [0.02, 0.11] (see Table 3).

### 3.4. Moderated Effect Test

Building upon the existing empirical findings that support the notion of an indirect relationship between social networking usage and emotional eating among adolescents, mediated by negative social comparison, we proceeded to investigate whether this indirect effect is moderated by the level of adolescent perspective-taking. To examine this phenomenon, we employed a conditional process model, specifically Model 7, utilizing the PROCESS macro.

The results displayed in Table 4 provide evidence of a significant interaction effect between social networking use and perspective-taking (B = −0.01, SE = 0.01, *p* < 0.05), indicating that perspective-taking moderates the relationship between social networking use and negative social comparison. Subsequent analyses were conducted to explore this interaction further. Specifically, when adolescents with high levels of perspective-taking engaged in social networking use, they still exhibited a certain degree of negative social comparison (B = 0.04, SE = 0.01, 95% CI = [0.02, 0.07]). However, their levels of negative social comparison were comparatively weaker than those observed among adolescents with moderate perspective-taking levels (B = 0.05, SE = 0.01, 95% CI = [0.03, 0.08]) and low perspective-taking levels (B = 0.06, SE = 0.01, 95% CI = [0.03, 0.08]) (see Figure 2). In summary, these findings robustly support the hypothesized moderating role of perspective-taking. As perspective-taking levels increase, adolescents are less likely to engage in pronounced negative social comparison when using social networking sites, thereby reducing the occurrence of emotional eating behaviors.

## 4. Discussion

Previous research has consistently shown that the use of social networking sites increases the risk of disordered eating behaviors [24,25]. However, the existing literature presents conflicting findings when it comes to the association between social media usage and emotional eating. While not directly examining social media usage, Caner et al. [59] conducted a survey among 1363 Turkish adolescents and found no significant link between social media addiction and emotional eating. They suggested that adolescent addiction to electronic games may have a stronger influence on emotional eating [60]. In contrast, Göbel et al. [61] discovered a possible positive correlation between social media addiction and emotional eating, as indicated by higher scores on the social media addiction scale among individuals exhibiting emotional eating tendencies. The present study directly measures adolescents’ social networking site usage to investigate its impact on emotional eating. Consistent with findings on disordered eating behaviors, this study provides evidence of a significant relationship between social media usage and emotional eating. It is important to note that the level of social media usage among the participating adolescents in this study was moderate rather than frequent. This result may be attributed to the fact that in Chinese secondary schools, teachers discourage students from using mobile phones during study hours, allowing limited usage only after school. Hence, this contextual factor contributes to the moderate level of social media usage observed among the adolescents in this study.

This study found that adolescents’ use of social networking sites is associated with increased engagement in negative social comparisons and that negative social comparison mediates the relationship between social media usage and emotional eating. However, it is worth noting that these findings appear to contradict previous research on the subject. Research has suggested that social media usage does not necessarily lead to negative outcomes, and the key factor lies in the patterns of social interaction and comparison on these platforms. If the social comparisons and interactions are positive, the use of social networking sites can bring benefits rather than harm [38,62]. Despite the potential benefits of positive social interactions, these may be hindered by the unique challenges and vulnerabilities faced by adolescents. Adolescents occupy a relatively vulnerable position within the broader social structure and face significant academic demands and stress. This puts them at a higher risk of engaging in upward comparisons with older individuals, which can contribute to a perception of personal inadequacy in terms of abilities. Additionally, the high academic pressure they experience depletes cognitive resources, potentially hindering their ability to engage in positive social interactions on social networks. Moreover, exposure to an abundance of positively biased content on social networking sites increases the likelihood of unfavorable social comparisons among adolescents [31,33]. Adolescents not only spend a considerable amount of time browsing content posted by other users on these platforms but also commonly encounter content that has a positive bias [26,32]. As a result, engaging in social media browsing is more likely to lead to negative social comparisons among adolescents, which, in turn, reinforces self-consciousness regarding their appearance, body shape, and abilities while generating lower self-evaluations. These negative evaluations and emotions further contribute to the adoption of emotional eating as a coping mechanism. Therefore, it is crucial to understand the complex interplay between social media usage, social comparisons, and emotional eating in order to provide appropriate support and interventions for adolescents facing these challenges.

While adolescents are prone to engaging in negative social comparisons on social networking sites, it is important to consider the protective factors that can mitigate these effects. Understanding these factors can help us better understand the impact of social media use on adolescents’ well-being. Research conducted by de Vries et al. found that happiness plays a moderating role in the relationship between Facebook use and social comparison. Specifically, individuals who are unhappier tend to experience stronger negative outcomes when using Facebook [63]. This suggests that happiness can act as a protective factor against the negative effects of social media on social comparisons. Similarly, a study conducted in China discovered that self-concept clarity can moderate the relationship between the use of WeChat Moments and negative social comparison. Among individuals with lower self-concept clarity, the use of WeChat Moments was associated with stronger negative social comparisons [37]. This highlights the importance of having a clear sense of self when engaging with social media platforms.

Another protective factor in social interactions is perspective-taking. It has been found to reduce personal stereotyping, enhance evaluations of out-group members, and decrease biases and discrimination [44,64,65,66,67]. In this study, we have discovered that perspective-taking plays a moderating role in the relationship between social media use and negative social comparisons. When individuals engage in higher levels of perspective-taking, they demonstrate a decreased tendency to engage in negative social comparisons. The process of social comparison is deeply intertwined with one’s self-concept, and perspective-taking serves to mitigate negative patterns of comparison via reducing the psychological distance between individuals and enhancing their self-esteem. These findings hold significant practical implications for alleviating the adverse impacts of social media on adolescents. Specifically, training adolescents in perspective-taking skills can prove instrumental in reducing emotional eating behaviors that stem from social networking usage. Through actively encouraging adolescents to consider and understand others’ viewpoints, interventions might foster the adoption of healthier coping strategies and promote positive psychological well-being within the realm of social media utilization.

Of course, there are certain limitations to this study. Firstly, emotional eating was measured through self-reporting by adolescents, and future research could employ additional measures from different sources, such as observations of actual eating behaviors. This would provide a more comprehensive understanding of the relationship between emotional eating and social media use. Secondly, it is important to recognize that the frequency and duration of social media use are not the only predictive variables when adolescents utilize social networking sites. Research has shown that the content browsed on these platforms plays a significant role in influencing individuals’ psychology and behavior. For example, exposure to weight-related content can lead to lower body satisfaction and increased frequencies of binge eating [28]. Moreover, content related to images on social networks tends to have a stronger influence on individuals [26,68]. Furthermore, it is essential to consider the dynamic nature of the relationship between social media content and eating behaviors. While this study adopts a static perspective in examining the link between social media use and emotional eating, in reality, the interaction between the content on social media and eating behaviors is more complex. For instance, adolescents may share their own emotional eating behaviors on social networking sites, which can subsequently impact both themselves and others. In summary, while this study provides valuable insights, it is important for future research to consider additional measurement methods and explore the dynamic nature of social media content and eating behaviors. Through doing so, we can gain a more nuanced understanding of the complex relationship between emotional eating and social media use. Lastly, it should be noted that the sample of the study primarily focused on adolescents in the middle school stage, which may not effectively represent the entire population of adolescents throughout the entire period of adolescence.

## 5. Conclusions

Heightened social networking usage fosters a greater propensity for negative social comparisons, subsequently amplifying the risk of emotional eating. Furthermore, it was observed that the perspective-taking skills of adolescents have the potential to effectively mitigate negative social comparisons and emotional eating behaviors associated with social media use.

## Figures and Tables

**Figure 1 behavsci-13-00768-f001:**
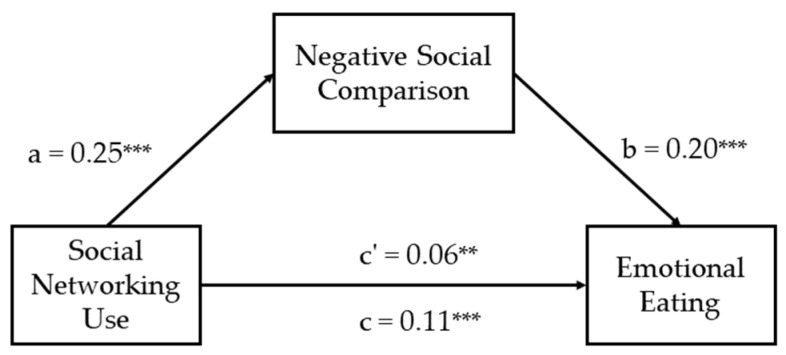
The mediation model with standardized estimates. ** *p* < 0.01, *** *p* < 0.001.

**Figure 2 behavsci-13-00768-f002:**
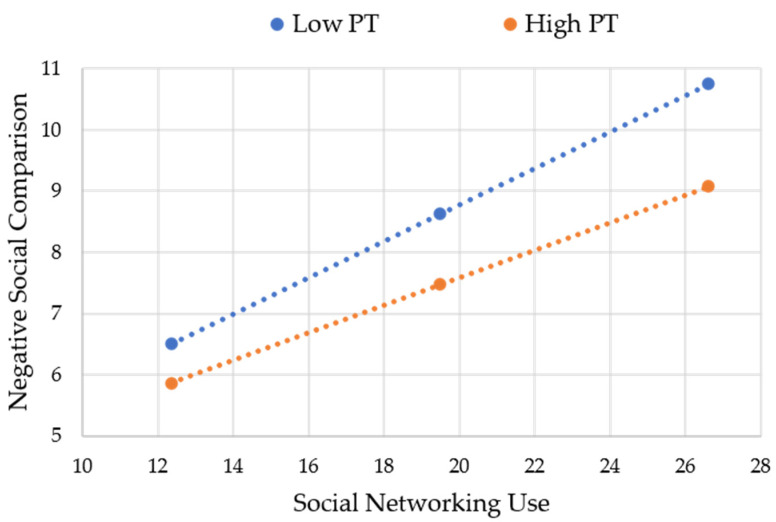
Moderation of indirect effect by perspective-taking.

**Table 1 behavsci-13-00768-t001:** Means, standard deviations, and correlations of the main study variables (N = 778).

	*M*	*SD*	1	2	3	4	5	6	7
1. Age	13.01	0.49	-						
2. Gender			−0.06	-					
3. BMI	24.97	6.16	0.12 **	0.004	-				
4. Social Networking Use	19.48	7.13	0.03	0.19 **	0.05	-			
5. Emotional Eating	8.70	4.04	0.06	0.07 *	0.06	0.21 **	-		
6. Negative Social Comparison	8.02	3.60	−0.06	0.13 **	0.03	0.50 **	0.23 **	-	
7. Perspective-taking	18.32	3.63	−0.08 *	0.04	−0.07	0.11 **	0.11 **	−0.09 *	-

Note: * *p* < 0.05, ** *p* < 0.01.

**Table 2 behavsci-13-00768-t002:** Testing the mediation effect of negative social comparison on the association between social networking use and emotional eating (N = 778).

Regression Equation		Fitting Index			Significance of Coefficients			
Outcome	Predictors	*R*	*R* ^2^	*F*	*B*	*t*	LLCI	ULCI
Emotional Eating	Gender	0.22	0.05	9.81	0.31	1.05	−0.26	0.88
	Age				0.40	1.34	−0.18	0.97
	BMI				0.03	1.13	−0.02	0.07
	Social Networking Use				0.11	5.48 ***	0.07	0.15
Negative Social Comparison	Gender	0.50	0.25	63.73	0.21	0.91	−0.24	0.66
	Age				−0.54	2.29 *	−0.99	−0.08
	BMI				0.01	0.57	−0.03	0.05
	Social Networking Use				0.25	5.30 ***	0.21	0.28
Emotional Eating	Gender	0.27	0.07	11.88	0.26	0.92	−0.30	0.83
	Age				0.50	1.72	−0.07	1.08
	BMI				0.02	1.05	−0.02	0.07
	Social Networking Use				0.06	2.75 **	0.02	0.11
	Negative Social Comparison				0.20	4.39 ***	0.11	0.29

Note: * *p* < 0.05, ** *p* < 0.01, *** *p* < 0.001.

**Table 3 behavsci-13-00768-t003:** Total, direct, and indirect effects of social networking use on emotional eating (N = 778).

	Effect Size	Boot SE	LLCI	ULCI	Relative Effect Value
Total effect	0.11	0.02	0.07	0.15	
Direct effect	0.06	0.02	0.01	0.02	56.50%
Indirect effect	0.05	0.01	0.02	0.07	43.50%

**Table 4 behavsci-13-00768-t004:** Testing the moderation effect of perspective-taking on the association between social networking use and social networking use (N = 778).

Regression Equation		Fitting Index			Significance of Coefficients			
Outcome	Predictors	*R*	*R* ^2^	*F*	*B*	*t*	*LLCI*	*ULCI*
Negative Social Comparison	Gender	0.53	0.28	49.07	0.19	0.86	−0.25	0.64
	Age				−0.60	2.59 **	−1.05	−0.14
	BMI				0.01	0.38	−0.03	0.04
	Social Networking Use				0.45	5.57 ***	0.29	0.60
	Perspective-taking				0.04	0.43	−0.13	0.20
	Social Networking Use × Perspective-taking				−0.01	2.43 *	−0.02	0.00

Note: * *p* < 0.05, ** *p* < 0.01, *** *p* < 0.001.

## Data Availability

The raw data supporting the conclusions of this article will be made available by the authors, without undue reservation.

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
