# Peer review of "Social Networking Site Use and Emotional Eating Behaviors among Chinese Adolescents: The Effects of Negative Social Comparisons and Perspective-Taking"

_behavsci, 2023, doi:10.3390/bs13090768_

Round 1

Reviewer 1 Report

Congratulations, you will find my comment in the file attached.

Yours sincerely.

The reviewer.

Author Response

Dear Reviewer,

Thank you for taking the time to review the manuscript titled [Social Networking Site Use and Emotional Eating Behaviors Among Chinese Adolescents: The Effects of Negative Social Comparisons and Perspective Taking]. We appreciate your expertise in the field and your thoughtful and constructive comments on the manuscript.

We have carefully considered your feedback and suggestions, and we would like to express our gratitude for your valuable contributions to improving the quality and rigor of our work. Your comments have been instrumental in strengthening the manuscript and ensuring that it meets the standards of our target academic audience.

We have made revisions to the manuscript based on your suggestions. The specific responses are provided below:

  1. The final sample included 778 participants, as data from 37 participants were excluded from the statistical analysis. The reasons for exclusion were severe missing values in their data or failure to pass the attention check. Specifically, an attention check question was added to the questionnaire, requiring participants to select "3" as the answer. Samples that did not select "3" did not pass the attention check.
  2. The TFEQ-R18 measures various eating behaviors, including cognitive restraint, disinhibition, and emotional eating. This questionnaire has been frequently used in Chinese samples, and Shi et al. (2011) demonstrated good reliability and validity of this questionnaire among Chinese female university students. The emotional eating dimension of the TFEQ-R18 consists of three items, the disinhibition dimension consists of nine items, and the cognitive restraint dimension consists of six items. Therefore, in this study, we utilized the three items from the emotional eating dimension of the TFEQ-R18 as our measurement tool.
  3. As per your suggestion, we have removed Figure 1 from the manuscript.
  4. According to your suggestion, we have incorporated content in the introduction section that discusses the influencing factors of emotional eating and its relationship with anxiety.
  5. We apologize for the excessive length of the title, as pointed out by the reviewer. However, we were unable to find a suitable way to shorten it and, therefore, no adjustments were made to the title in the manuscript.

Reviewer 2 Report

The manuscript “Social Networking Site Use and Emotional Eating Behaviors Among Chinese Adolescents: The Effects of Negative Social Comparisons and Perspective Taking” commendably addresses a timely and relevant topic, providing a holistic view of the relationship between social networking use and emotional eating among adolescents. The application of the bias-corrected percentile bootstrap method and the PROCESS macro to test mediation and moderated effects is noteworthy. This demonstrates a rigorous analytical approach, enhancing the validity of the findings. The correlation and mediation results are well-structured and succinctly presented, making it straightforward for readers to grasp the primary outcomes of the research. The authors deserve praise for highlighting the unique contextual factors in Chinese secondary schools that might influence social media usage. This addition provides a richer understanding of the study's environment. The effort to check and report on potential biases, including common method bias and non-response bias, is commendable. It provides transparency and increases confidence in the study's findings. The discussion section adeptly connects the study's findings with existing literature, providing both agreement and contrast, which enriches the overall narrative of the research. The study's hypotheses are clearly laid out, making it easier for readers to follow the logic and flow of the research. The authors successfully tap into a modern-day concern – the influence of social networking sites on adolescents' well-being – which is of great interest to both academics and the general public. The conclusion effectively distills the study's primary insights, offering readers a clear and concise summary of the research's significance and implications.

Overall, the manuscript is a well-conducted piece of research that offers valuable insights into an area of increasing societal concern. The authors have undertaken a rigorous analytical approach and provided a comprehensive understanding of the subject.

Even though there are several concerns need addressing:

1.       The manuscript, while providing some context about the relationship between social networking sites and emotional eating, could benefit from a more detailed literature review. Specifically, the authors might consider discussing prior theoretical frameworks or models that explain this relationship.

2.       The choice of Harman's single-factor test is appreciated to address common method bias. However, this test has been critiqued for its oversimplification. The authors might consider complementing this with other methods to ensure comprehensive bias handling.

3.       The study seems to emphasize correlations between certain variables like social media use and emotional eating. The authors need to be cautious in how they present these findings, ensuring they don’t inadvertently imply causation where only correlation has been demonstrated.

4.       The study context is confined to Chinese secondary schools. While it is understood that every study has its geographic and demographic limitations, the authors should elaborate more on whether their findings can be generalized beyond this group.

5.       The term "perspective-taking" needs clearer operationalization. How exactly was this measured, and how do the authors define "high", "medium", and "low" levels of perspective-taking?

6.       While the manuscript does provide statistical evidence supporting its claims, the effect sizes, in many cases, seem modest. It would be beneficial if the authors could discuss the practical significance of these findings in addition to the statistical significance.

7.       In the implications section, the authors suggest interventions in schools and homes. It would add depth if the manuscript could propose what these interventions might look like based on the study’s findings.

 8.       The discussion touches upon conflicting findings in the literature but does not delve deep into reconciling or explaining these discrepancies. The authors might consider offering more insights into why their findings might differ from past research.

9.       The conclusion section, while informative, seems a bit repetitive, reiterating many points already mentioned in the discussion. The authors could streamline this section to offer succinct takeaways from the study without redundancy.

Minor editing is needed.

Author Response

Dear Reviewer,

Thank you for taking the time to review the manuscript titled [Social Networking Site Use and Emotional Eating Behaviors Among Chinese Adolescents: The Effects of Negative Social Comparisons and Perspective Taking]. We appreciate your expertise in the field and your thoughtful and constructive comments on the manuscript.

We have carefully considered your feedback and suggestions, and we would like to express our gratitude for your valuable contributions to improving the quality and rigor of our work. Your comments have been instrumental in strengthening the manuscript and ensuring that it meets the standards of our target academic audience.

We have made revisions to the manuscript based on your suggestions. The specific responses are provided below:

  1. We have added a section on using single unmeasured latent method to test for common method bias.
  2. We have modified expressions that suggest causality too strongly.
  3. Perspective taking (PT) is defined as the tendency to consider issues from other people's perspectives. In this study, PT was measured using four items that have been used in previous research, including "I look at things from the perspective of others," "I imagine how my actions will affect things that are important to others," "I understand why particular issues hold emotional significance for others," and "I look at matters in terms of other people's personal concerns." In testing for moderation effects, scores that deviated from the mean by plus or minus one standard deviation were considered high PT levels and low PT levels, respectively.
  4. We have added a discussion on the practical implications and potential interventions.
  5. We have added an analysis of inconsistencies in previous studies in the discussion section.
  6. We have simplified the conclusions section.
  7. We have added a discussion on whether the sample of high school students can be generalized to other samples in the discussion section.
  8. We have added content on the relationship between online behavior and emotional eating to the literature review section.

Reviewer 3 Report

The work presented by the authors addresses an interesting and growing problem among the young population, such as the case of emotional hunger and the use of social networks. Although the topic is relevant, the following points should be taken into account in order to improve the quality of the work and thus allow its publication to be considered.

1.                  Line 27 gives a definition of emotional eating that includes the negative aspect of the emotion. Line 32 speaks of negative emotional eating as that which is produced to regulate negative emotions. If the definition of emotional eating already includes the negative aspect of the emotions to be regulated, then the concept should be used as it is, without adding emotional valence. If not, the definition should be modified.

2.                  Add the acronym SNS on line 45 or delete it from the summary.

3.                  Revise the text on lines 116-119 regarding self-esteem.

4.                  Specify the platform on which the online data were collected. Also explain in more detail how the sample was selected (type of sampling, recruitment, etc.). All of this should be included in the procedure subsection.

5.                  Line 146: 2 decimals for age and standard deviation.

6.                  Revise line 154, Questionnaire name?

7.                  Line 163 2 decimal places for alpha. Do the same for the rest of the values. Revise the whole text and homogenise the use of decimals.

8.                  It is not specified what kind of data analysis has been carried out. Use the data analysis subsection.

9.                  Nothing is said about the ethical aspects of the research.

10.              If, as stated in line 161, the participants were only asked about the use of WeChat Moments, this should be specified instead of using the SNS concept in a global way. The whole paper should therefore be revised to specify this question. An explanation of what the app is and how it works should also be provided for non-app users.

11.              Revise the footer of Table 1. The gender does not make sense.

12.              Give effect sizes.

13.              Give the confidence interval with limits in the individual reports of the effects of the variables.

14.              It is recommended to include a table with the results of the mediation analysis, indicating the direct and indirect effects.

15.              A figure showing the standardised regression coefficients of the mediation model should be included.

16.              A figure showing the unstandardised simple slopes for reporting moderation is recommended.

Author Response

Dear Reviewer,

Thank you for taking the time to review the manuscript titled [Social Networking Site Use and Emotional Eating Behaviors Among Chinese Adolescents: The Effects of Negative Social Comparisons and Perspective Taking]. We appreciate your expertise in the field and your thoughtful and constructive comments on the manuscript.

We have carefully considered your feedback and suggestions, and we would like to express our gratitude for your valuable contributions to improving the quality and rigor of our work. Your comments have been instrumental in strengthening the manuscript and ensuring that it meets the standards of our target academic audience.

We have made revisions to the manuscript based on your suggestions. The specific responses are provided below:

  1. Based on the reviewer's suggestion, the term "negative" has been removed from Line 32.
  2. The acronym SNS has been added to Line 45.
  3. Modifications have been made to the content in lines 116-119.
  4. The platforms for online data collection have been explained in the Participants section.
  5. The questionnaire name has been added as the Social Networking Usage Intensity Scale.
  6. All data are reported with two decimal places.
  7. Effect sizes and confidence intervals for the mediating effects have been included in the Results section.
  8. Information regarding the ethical review of the study has been added.
  9. An introduction to WeChat Moments (a social networking site) has been included. Among the participants surveyed, WeChat Moments was identified as the most commonly used social media platform. Therefore, the intensity of WeChat Moments usage serves as an indicator of social media usage in this study.
  10. Figures and tables depicting the mediation and moderation effects have been added to the Results section.